# Motion Planning in Compressed Representation Spaces

**Lukas Lao Beyer** [1]  **Sertac Karaman** [1]

## Abstract

Deep learning methods have vastly expanded the capabilities of motion planning in robotics applications, as learning priors from large-scale data has been shown to be essential in capturing the highly complex behavior required for solving tasks such as manipulation or navigation for autonomous vehicles. At the same time, model-based planning algorithms based on search or optimization remain an essential tool due to their flexibility, efficiency, and the ability to incorporate domain knowledge via expert-designed algorithms and objective functions. We propose a new generative framework to unify these two paradigms. First, we learn an autoencoder with a high compression ratio and a latent space of hierarchically ordered, discrete-valued tokens. Leveraging both the dimensionality reduction and the hierarchical coarse-to-fine structure learned by this autoencoder, we then perform motion planning by directly searching in the latent space of tokens. This search can optimize arbitrary objective functions specified at test time, providing a large degree of flexibility while maintaining efficiency and producing realistic solutions by relying on the generative capabilities of the highly compressed autoencoder. We evaluate our method on nuPlan and the Waymo Open Motion Dataset, showing how latent space search can be used for a variety of guided behavior generation tasks, achieving strong performance for closed-loop motion planning and multi-agent guided scenario synthesis without requiring any task-specific training.

## 1. Introduction

In real-world settings, robotic motion planning must produce trajectories that satisfy task- and user-specific objec-

[1]MIT LIDS. Correspondence to: Lukas Lao Beyer <llb@mit.edu>.

*Proceedings of the 43rd International Conference on Machine Learning*, Seoul, South Korea. PMLR 306, 2026. Copyright 2026 by the author(s).

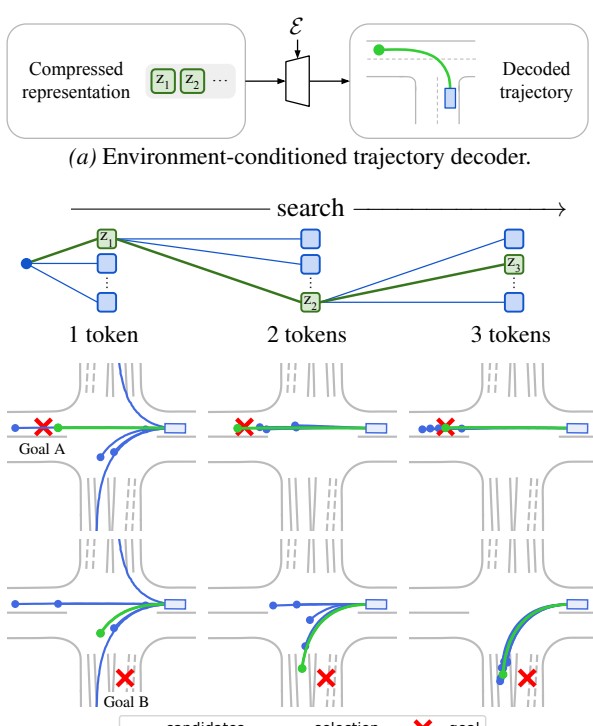

*(a)* Environment-conditioned trajectory decoder.

*(b)* Latent token search with test-time objective (no retraining).

*Figure 1.* **Token search for guided trajectory synthesis.** We learn a highly compressed, environment-conditioned autoencoder that maps trajectories to an ordered discrete code $z$. At test time we perform greedy token search: at step $i$, we fix $z_{1:i-1}$, enumerate $z_i$, decode each candidate trajectory, score according to the user-specified objective *chosen at test time*, and keep the best prefix (here we illustrate a goal-reaching objective). Because tokens are causally ordered and learned with nested dropout, each additional token refines a feasible prefix so greedy search follows the representation's intended coarse-to-fine semantics. Token semantics are not hand-designed and emerge through compression.

tives while remaining realistic and safe in complex environments. Classical model-based planners offer flexibility through explicit cost functions and constraints, and can be highly robust and performant when deployed in controlled settings (Foehn et al., 2021; Moore et al., 2014; Goh & Gerdes, 2016). However, purely model-based approaches struggle in unstructured and open-ended domains, where realistic behavior depends on rich priors learned from data. This has driven a shift toward learning-based approaches that leverage large-scale data to acquire such priors. However, many learning-based planners bake task structure into

the model via conditioning (Cheng et al., 2024), training-time reward shaping (Zhang et al., 2025), or specialized inference procedures (Dauner et al., 2023), which can make it nontrivial to accommodate new objectives or constraints.

In this paper, we propose a generative framework aimed at unifying these paradigms by separating trajectory realism from test-time control. A key ingredient enabling this separation is compression. Consider mapping high-dimensional trajectories to short sequences of low-dimensional or even discrete tokens: this simultaneously (i) reduces the dimensionality of the decision space the planner must search over and (ii) concentrates the burden of producing realistic motion in a decoder, i.e., a learned reconstruction model that expands tokens back into full trajectories. Figure 1 previews our approach, which implements this principle by learning an environment-conditioned autoencoder that maps trajectories to a highly compressed, ordered latent representation; motion planning then reduces to tree search over latent tokens guided by a user-specified test-time objective. This yields a simple interface for incorporating domain knowledge or preferences while relying on the decoder to produce realistic, context-appropriate behavior.

From a motion-planning viewpoint, the environment-conditioned, ordered tokenization induces a context-dependent library of maneuvers. Unlike classical maneuver libraries with a small fixed set of hand-engineered primitives (Frazzoli et al., 2005; Pivtoraiko et al., 2009), the library here is learned from data and automatically adapts to the scene through environment conditioning. Furthermore, the hierarchical ordering of this library enables efficient traversal of a combinatorially large set of maneuvers via coarse-to-fine refinement.

From the point of view of guided generation, our method chooses a different route than the iterative gradient-based guidance commonly used with diffusion and flow models, offering unique advantages. For example, partial token prefixes decode to trajectories in the original state space, so the objective can be applied directly without requiring additional estimators. Furthermore, latent token search naturally supports objectives that are not smooth or differentiable.

We evaluate our approach on the Waymo Open Motion Dataset (Ettinger et al., 2021) and nuPlan (Caesar et al., 2021), using autonomous driving as a demanding testbed for guided generation with diverse objectives. Overall, our results indicate that a highly compressed, ordered latent representation can turn guided generation into a direct search problem. Throughout our experiments, we find that greedy search in the learned token space provides a simple yet effective mechanism for test-time control, supporting a variety of objectives, yielding strong performance on closed-loop planning, and achieving high realism metrics on guided multi-agent scenario synthesis, all without task-specific training.

## 2. Related Work

**Test-time guidance for generative models.** A large body of work studies how to steer generative models at test time. In diffusion models (Ho et al., 2020), guidance is often implemented by modifying the sampling dynamics using gradients of a conditioning signal, including classifier guidance (Dhariwal & Nichol, 2021), classifier-free guidance (Ho & Salimans, 2021), and guidance using more general functions (Song et al., 2023; Bansal et al., 2023). These techniques have also been adopted for trajectory generation in robotics, e.g., Diffusion Policy (Chi et al., 2023), and for controllable motion prediction in multi-agent settings, e.g., MotionDiffuser (Jiang et al., 2023a). Our work differs in the mechanism used to realize test-time guidance: instead of gradient-based steering during sampling, we perform decode-and-score search over ordered latent tokens, which naturally supports objectives that are non-differentiable and can be evaluated directly on decoded trajectories even in intermediate search iterations.

**Tokenization and ordered representations.** Tokenization via autoencoders is a standard technique for reducing the dimensionality of generative modeling problems. In image modeling, generators commonly operate in learned latent spaces of continuous VAEs (Kingma & Welling, 2014) or discrete tokenizers such as VQ-VAE/VQGAN (van den Oord et al., 2017; Esser et al., 2021) to improve efficiency and scalability (Rombach et al., 2022; Chang et al., 2022; Li et al., 2024). Recently, nested dropout (Rippel et al., 2014) has been used to learn ordered and variable-length image tokenizations, inducing a coarse-to-fine hierarchy in which shorter codes capture global structure and longer codes add detail (Wen et al., 2025; Miwa et al., 2025; Bachmann et al., 2025). We build on this perspective by learning a hierarchical trajectory representation. However, unlike image tokenizers, our trajectory tokenizer is environment-conditioned, which shifts part of the conditional modeling burden into the decoder.

**Training-free generation in tokenizer latent spaces.** Training-free test-time generation has been explored by optimizing latent codes of pretrained generators. Examples include optimizing GAN latents (Goodfellow et al., 2014) with CLIP-based objectives (Radford et al., 2021; Patashnik et al., 2021) and optimizing discrete tokenizer codes as in VQGAN-CLIP (Crowson et al., 2022). While such methods have historically been most effective for image editing, recent work suggests that extremely high compression tokenizers like TiTok (Yu et al., 2024) can make lightweight test-time latent manipulation surprisingly effective for generation (Lao Beyer et al., 2025). Our work is inspired by the observation that extreme compression can shift generation complexity toward the decoder, but differs in the inference mechanism: we leverage ordering and discretiza-

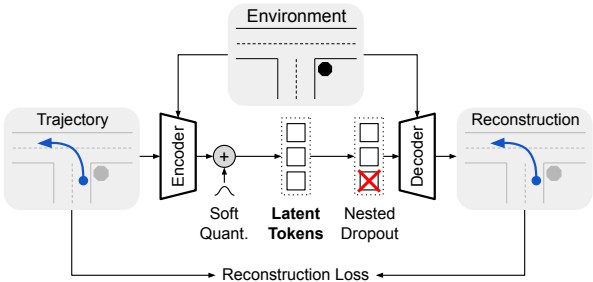

*Figure 2.* **Conditional autoencoder training.** A conditional autoencoder is trained with a reconstruction objective to capture a highly compact latent representation of the input trajectory given a particular environment. During training, we adaptively inject noise to the latent tokens as a form of soft quantization, and apply causal masking and nested dropout to learn an ordered representation.

tion to perform tree search over tokens rather than using continuous gradient-based latent optimization. Separately, compact learned latents have also been used for control by planning over latent actions with learned dynamics using tree search (Ozair et al., 2021; Jiang et al., 2023b). While algorithmically adjacent in their use of search over learned discrete representations, these methods target latent action planning rather than objective-guided trajectory generation, and are not directly comparable to our setting.

## 3. Conditional Trajectory Autoencoder

Trajectory prediction and planning problems in robotics must consider information such as sensor inputs or known maps. We therefore choose to represent a trajectory *conditionally* on given information about the environment and design a conditional autoencoder with the goal of learning a compact and expressive latent representation of the trajectory under a particular fixed environment (Figure 2).

**Notation.** We denote the environment as $\mathcal{E}$. In the typical autonomous driving setting, $\mathcal{E}$ consists of static world features such as road edges, road lines, lane geometry, stop signs and traffic lights, as well as dynamic object history for each visible agent in the scene. The full trajectory of the agent of interest is denoted as $\mathcal{T}$. The encoder Enc then produces a sequence of $N$ $D$-dimensional latent tokens $z := \{\mathbf{z}_i \in \mathbb{R}^D\}_{i=1}^N = \texttt{Enc}(\mathcal{T}, \mathcal{E})$ from the full trajectory $\mathcal{T}$ of the agent of interest and the environment $\mathcal{E}$. The decoder Dec attempts to reconstruct the trajectory $\mathcal{T}$ from the latent tokens and the environment, producing a prediction $\mathcal{T}_{\text{pred}} = \texttt{Dec}(\mathbf{z}, \mathcal{E})$. We represent trajectories as sequences of position samples uniformly spaced in time.

**Prediction space and loss function.** The decoder Dec predicts mean and variance parameters of a Gaussian distribution for each point along the trajectory, and is trained by minimizing the negative log-likelihood (NLL) of the ground truth reconstruction under the prediction. In practice, we make use of the $\beta$-NLL (Seitzer et al., 2022).

### 3.1. Compression via Adaptive Soft Quantization

To mitigate training challenges commonly associated with vector quantization while maintaining its desirable regularizing effects and enabling discrete search at test time, we use a form of *soft* quantization consisting of noise injection at the autoencoder bottleneck:

$$\texttt{corrupt}(\mathbf{z}) = \tanh(\mathbf{z}) + \boldsymbol{\epsilon}_t \qquad (1)$$

with $\boldsymbol{\epsilon}_t \sim \mathcal{N}(0, I\sigma_t^2)$. Note that a $\tanh$ activation is applied pointwise to the input, effectively creating an amplitude-limited noisy channel.[1] The chosen noise level $\sigma_t^2$ is picked adaptively during training to gradually ramp up from zero until a desired reconstruction accuracy is achieved. During training, we apply corrupt to each token $\mathbf{z}_i$ before feeding it to the decoder. At test time, we set $\sigma_t = 0$. We find that this adaptive noise schedule outperforms choosing a fixed noise level (further details can be found in Section A.4).

**Hard quantization at test time.** At test time, we leverage the decoder's ability to predict starting from heavily noised input tokens by quantizing its input. For this purpose, we round each token elementwise to the nearest discrete level, out of $N_l$ uniformly spaced levels. While reminiscent of finite scalar quantization (FSQ) (Mentzer et al., 2024), we apply hard quantization at test time and not during training.

### 3.2. Variable-Length Latent Encoding

To allow flexible reconstruction fidelity at test time and to facilitate structured exploration of the latent space, we choose to impose a causal ordering structure on the latent tokens $\mathbf{z}_i$. Causality among the tokens $\mathbf{z}_i$ is enforced via masked self-attention in the encoder and decoder networks. We additionally make use of nested dropout (Rippel et al., 2014) to allow the decoder to operate on variable-length encodings. During training, nested dropout drops a random number of tail tokens from the encoding, forcing the decoder to reconstruct the trajectory from a truncated code.

### 3.3. Network Architecture

Our conditional autoencoder is implemented using three transformer models: an environment encoder which processes static world and dynamic object history information, and encoder and decoder models which attend to the environment information via cross-attention and process the latent tokens via causal self-attention. Our environment encoder follows Motion Transformer (MTR), making use of MTR's local neighborhood attention (Shi et al., 2022). At the bottleneck, we project transformer tokens to the de-

---

[1]We refer to this process as soft *quantization* since our corrupt procedure resembles an amplitude-limited Gaussian channel, for which the input distribution achieving maximum information capacity is discrete (Smith, 1971).

sired low dimensionality $D$ and during training apply nested dropout and noise corruption for soft quantization.

**Multi-agent modeling.** Joint tokenization of several agents' trajectories offers potential for learning representations that exploit the correlations between trajectories present in multi-agent interactions. The environment encoder and cross-attention blocks are unchanged compared to the single-agent model, and process each agent's trajectory individually in the agent-centric coordinate frame. To jointly model all agents, we use self-attention blocks with a hybrid attention mask allowing each agent trajectory token to attend to the other agent trajectories bidirectionally, while latent tokens can aggregate information from the set of all agents. As before, causal masking is imposed among latent tokens.

Section A.1 provides additional network architecture details.

## 4. Latent Token Search

Causal masking combined with nested dropout ensures that later tokens capture increasingly more fine-grained information about the encoded trajectory. Therefore, at test time, latent token sequences of any size may be used to obtain reconstructions of varying degrees of fidelity. For planning tasks, this property suggests a *greedy* latent space search strategy in which tokens can be picked one after another.

More precisely, consider evaluating the trajectory $\mathcal{T}_{\text{pred}}$ according to some objective function $f(\mathcal{T}_{\text{pred}}) \mapsto (C, \leq_C)$ specified at test-time. Note that the output $C$ can be any set admitting a total order $\leq_C$ (e.g., $f$ can be a scalar cost or a lexicographically ordered tuple, as in Section 5.3). We seek to find the trajectory that minimizes this objective by exploring the latent space of our decoder as follows. Starting with the first token $\mathbf{z}_1$, we enumerate its possible quantized values to find the assignment minimizing the cost of the decoded trajectory $f(\text{Dec}([\mathbf{z}_1], \mathcal{E}))$. Tokens are intentionally low-dimensional and heavily quantized, so enumeration is cheap (we score $N_l^D$ candidates per step). We iteratively repeat this process for the next token while keeping the previous token assignments fixed. Figure 1 illustrates this procedure, visualizing how successively selected tokens progressively refine a trajectory under a goal-reaching objective.

**Variance prediction for out-of-distribution detection.** Recall that our decoder is trained to predict the parameters of a Gaussian distribution for each point along the trajectory. We find that the predicted variance is correlated with the reconstruction error (see Section A.5 for more details). Therefore, an effective strategy to detect and discard decoded trajectories of poor quality is to penalize high-variance predictions.

**Optional continuous refinement.** The decoder can also be evaluated on continuous latents, enabling optional gradient-based refinement of the tokens selected by discrete search,

either for differentiable task objectives or auxiliary criteria such as predicted variance reduction. Discrete search remains essential, as continuous optimization from scratch performs substantially worse. See Section A.7 for details.

## 5. Experiments

In our experiments, we answer the following three questions.

**Reconstruction:** *does the learned encoding cover the space of trajectories sufficiently?* In Section 5.1 we verify that this is the case by analyzing the autoencoder's reconstruction accuracy. We also evaluate the ability of greedy token search to perform reconstruction, validating the effectiveness of our ordered latent representation.

**Semantics:** *do encodings possess a high-level semantic interpretation?* In Section 5.2 we qualitatively show that behavior can be transferred between different scenarios by simply copying tokens, indicating that certain encodings can be interpreted as mapping to certain high-level behavior.

**Guidance:** *is token search adaptable to arbitrary objective functions, other than reconstruction?* In Sections 5.3 and 5.4 we present results of behavior synthesis according to a variety of objectives, including a realistic route-following objective that enables the deployment of our framework as a self-driving planner in closed-loop.

In answering these questions, we consider two different datasets: the Waymo Open Motion Dataset (WOMD) (Ettinger et al., 2021) for open-loop experiments, and nuPlan (Caesar et al., 2021) for closed-loop evaluation. We also consider two different domains: *marginal* single-agent as well as *joint* multi-agent trajectory modeling.

Finally, we ablate the main representation learning design choices in Section 5.5, finding that our proposed soft quantization and the use of a very low-dimensional bottleneck are key to the success of our approach.

### 5.1. Reconstruction

**Hard quantization at test-time.** Before attempting discrete tree search over quantized tokens, we analyze reconstruction accuracy under hard quantization on the WOMD validation set. As shown in Table 1, we find that the conditional autoencoder's drop in reconstruction accuracy is relatively modest even when heavily quantizing the tokens to $N_l = 2$.

**Greedy search.** To check whether greedy token search is a suitable strategy for exploring the space of latent tokens, we consider a reconstruction objective that minimizes the average displacement error (ADE) of all position samples along the trajectory with respect to the ground truth. The greedy search with reconstruction objective forms a valid replacement for the learned encoder in our autoencoder.

*Table 1.* **Reconstruction under quantization and with search.** Greedy search in quantized space can match or exceed the reconstruction performance of the learned encoder. ADE metric is averaged over object types and prediction horizons following WOMD convention. $N_l$ denotes number of discretization levels.

| Num. tokens | Average Displacement Error (ADE) ↓ | | | | |
| --- | --- | --- | --- | --- | --- |
| | Autoencoder | | | Greedy Search | |
| | $N_l = 2$ | $N_l = 3$ | cont. | $N_l = 2$ | $N_l = 3$ |
| 1 | 0.800 | 0.617 | *0.567* | 0.708 | **0.524** |
| 2 | 0.519 | 0.410 | *0.365* | 0.485 | **0.363** |
| 3 | 0.403 | 0.334 | **0.298** | 0.386 | *0.301* |

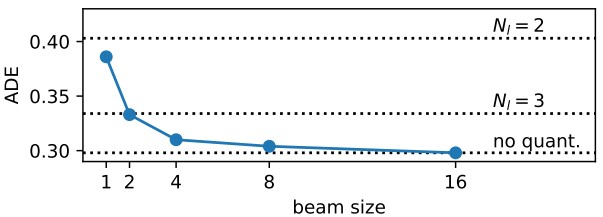

*Figure 3.* **Increasing beam size in latent token search with reconstruction objective has diminishing returns.** Thanks to the hierarchical structure of the learned latent space, greedy search is already an effective search strategy for reconstruction. (Blue trace: beam search results with $N_l = 2$; dashed lines: autoencoder reconstructions with and without quantization.)

Indeed, Table 1 shows that greedy search significantly outperforms the learned encoder, demonstrating that greedy token selection is a valid approach thanks to the causal and noise-resilient structure of the autoencoder's latent space.

**Beam search.** We find that using a less greedy search algorithm leads only to marginal improvements to reconstruction accuracy. This is demonstrated in Figure 3 using beam search, which keeps a set of top candidates at each iteration, rather than just a single one.

Overall, we find the latent space of our decoder is sufficiently expressive using just three tokens, each heavily quantized to only three bits. This is evidenced by a reconstruction ADE that is significantly lower than state-of-the-art motion prediction models. For example, DriveGPT achieves a minimum ADE of 0.524 across 6 predictions (Huang et al., 2025a), while our decoder is able to reconstruct the ground truth with an ADE of 0.386. We also find that our proposed representation can be efficiently traversed via greedy search, enabling our latent space to represent a large library of trajectories with an efficient hierarchical structure.

### 5.2. Semantics

We qualitatively show that latent token encodings learned by our conditional autoencoder carry high-level semantic information through *token swapping*. Concretely, consider decoding the latent token representation of a given trajectory $\mathcal{T}_A$

Tokens copied from...

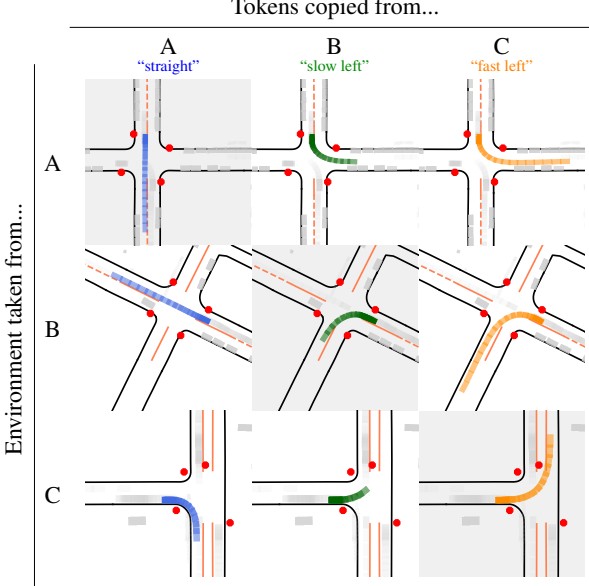

*(a)* Swapping encodings between scenarios

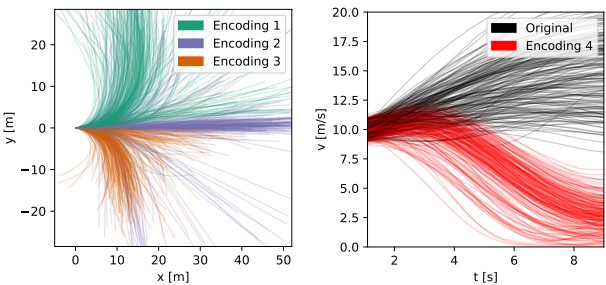

*(b)* Tokens repeatedly transfer behavior

*Figure 4.* **Tokens have meaningful environment-dependent semantics.** When copying the encoding of a given trajectory under a particular environment of the WOMD test set and decoding it under a *different* test environment (Equation (2)), predictable behavior consistent with the new environment is produced. In *(a)*, Shaded plots show the reference trajectory reconstructed in its original environment, while the remaining plots correspond to decoding the trajectory's encoding in a different environment. Note that Environment C does not admit driving straight, and the decoder produces a valid alternative reconstruction (last row, first column). In *(b)*, we decode each latent token encoding from a small pre-selected set (Encoding 1, 2 or 3) in many environments containing intersections (left). We also select an additional Encoding 4 corresponding to a high deceleration event, and plot the speed profiles resulting from decoding it in many different environments alongside the original speed profiles from those environments (right).

in its environment $\mathcal{E}_A$ under a *different* environment $\mathcal{E}_B$:

$$\mathcal{T}_{A \to B} = \text{Dec}(z_A, \mathcal{E}_B) \quad \text{with} \quad z_A = \text{Enc}(\mathcal{T}_A, \mathcal{E}_A). \quad (2)$$

Figure 4a shows the result of this simple manipulation for selected scenarios where the agent of interest is a vehicle at an intersection. We observe that decoding an encoding corresponding to a desired behavior in a different scenario can be used to transfer this behavior to a novel scenario.

Furthermore, we observe that token swapping can result in

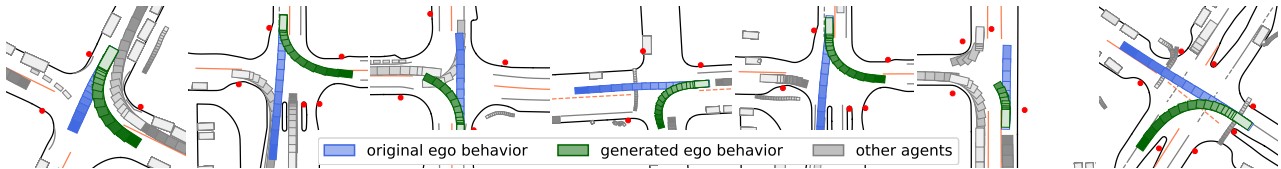

*Figure 5.* **Guided trajectory generation examples.** Token search synthesizes desired behavior (left turn; green) according to a test-time user-defined objective (left turn, in this example). Compared to the original behavior recorded in the ground truth (straight; blue), the synthesized trajectory is novel in order to comply with the left-turning objective.

*Table 2.* **Maneuver optimization with left-turn objective.** Greedy token search efficiently explores the conditional autoencoder's latent space with increasing success rate for increased search depth, while remaining robust against infeasible guidance in cases where the maneuver would be invalid. *Infeasibility* refers to kinematic infeasibility of the synthesized trajectory.

| | Did turn? | Edge contact | Infeasibility |
|---|---|---|---|
| *Scenarios with legal left turn available (234 cases):* | | | |
| Ground truth | 0% | 0.43% | 0% |
| Greedy (1 token) | 68.4% | 0.85% | 0% |
| Greedy (2 tokens) | 88.0% | 1.63% | 1.71% |
| Greedy (3 tokens) | 90.2% | 0.85% | 1.28% |
| Beam4 (3 tokens) | 88.9% | 0.43% | 2.13% |
| *Scenarios without available left turn (42 cases):* | | | |
| Ground truth | 0% | 0% | 0% |
| Greedy (1 token) | 0% | 2.38% | 0% |
| Greedy (2 tokens) | 0% | 2.38% | 0% |
| Greedy (3 tokens) | 2.38% | 2.38% | 0% |
| Beam4 (3 tokens) | 9.52% | 2.38% | 2.38% |

*Table 3.* **Maneuver optimization with lane-change objective.** We again find increasing success rate for increased search depth with a non-differentiable, lexicographic lane change objective that first enforces the desired number of lane changes and then minimizes lane deviation. *LC success* refers to executing the desired number of lane changes.

| | LC success | Center dist. | Avg. speed |
|---|---|---|---|
| *Objective: zero lane changes* | | | |
| Ground truth | 0% | 0.629 | 17.14 |
| Greedy (1 token) | 100% | 0.287 | 16.22 |
| Greedy (2 tokens) | 100% | 0.229 | 16.05 |
| Greedy (3 tokens) | 100% | 0.218 | 15.96 |
| Beam4 (3 tokens) | 100% | 0.205 | 15.88 |
| *Objective: one lane change* | | | |
| Ground truth | 100% | 0.629 | 17.14 |
| Greedy (1 token) | 53.85% | 0.669 | 18.28 |
| Greedy (2 tokens) | 97.60% | 0.700 | 17.92 |
| Greedy (3 tokens) | 99.04% | 0.670 | 17.75 |
| Beam4 (3 tokens) | 100% | 0.630 | 17.18 |

highly repeatable behavior transfer. Figure 4b shows two examples: (i) we decode just three different encodings (corresponding to turn directions at an intersection) in many different intersection scenarios, and (ii) we decode a single encoding corresponding to harsh deceleration, again in many different environments. In both cases, decoding the reference encoding in novel environments reliably results in the desired behavior being transferred, suggesting a class of maneuvers may be characterized by a particular token sequence in an interpretable way.

### 5.3. Guidance

Token swapping offers a simple, single-step editing primitive, but it cannot in general optimize an arbitrary objective or incorporate complex preferences. In this section we therefore use guided search over tokens to produce trajectories that satisfy a desired behavior. We begin with a detailed analysis of open-loop maneuver editing performance according to two separate objectives (Tables 2 and 3) and conclude with closed-loop planning results on the closed-loop nuPlan benchmark (Table 4). The experiments in this section make use of the variance-based out-of-distribution detection described at the end of Section 4.

**Maneuver editing.** We synthesize maneuvers according to the following objectives.

*Left-turn objective* — This objective directly maximizes the cumulative leftward heading change along the trajectory and penalizes excessive curvature. Since it does not take into account road geometry or obstacles at all and completely lacks realism or safety components, this objective is unsuitable for use in a traditional trajectory optimization framework.

*Lane-change objective* — We design a lexicographically ordered lane-change objective with a non-differentiable component. The objective first counts the integer number of lane changes as detected by changes in which lane center is closest. It then computes a lane-centering component which measures the deviation from the closest lane's centerline. The minimization target is the tuple containing the (integer) count of excess or missing lane changes followed by the (real-valued) lane-centering residual.

Table 2 presents the results of the *left turn* experiment, which demonstrates that token search with the simple left turn objective is able to achieve a high rate of success in cases where a left turn is indeed available, while remaining robust against invalid guidance provided by this objective in cases where a left turn is not available. For this evaluation, we filter the WOMD validation set to collect scenarios with the ego vehicle traversing an all-way stop intersection in the straight direction. We then subdivide these scenarios into (i) a subset where a legal left turn is possible and (ii) a

subset in which a left turn is not possible. Success rate[2] is significantly improved with increasing search depth, demonstrating the importance of maintaining a large library of maneuvers as well as the effectiveness of greedy search in efficiently traversing it. Furthermore, note that the decoder automatically ensures that behavior is consistent with the given scenario as evidenced by near-zero rates of contact between the predicted trajectory and road edge geometry, and near-zero rates of left-turning in scenarios without an available left turn. Example scenarios showing the original and newly synthesized behavior can be found in Figure 5.

Table 3 presents the results of the *lane-change* experiment. Full-depth greedy search yields near-perfect compliance with the desired number of lane changes while minimizing lane center deviation, demonstrating that latent token search can effectively optimize non-differentiable, structured objectives. We select 208 WOMD validation scenarios with a ground-truth left lane change. In Table 3 we report average speed as a diagnostic signal and to confirm forward progress. The filtered scenarios often correspond to overtaking maneuvers and we observe that the generated speed profiles are consistent with slower driving when suppressing a lane change, while matching the ground-truth speed otherwise.

Throughout our experiments, we also include beam search with a beam size of four (*Beam4* in Tables 2 and 3). While Beam4 yields a small improvement in lane-centering in our lane-change experiments, the left-turn experiment shows that broader search can *reduce* validity, e.g., by increasing left-turn rates in scenes where a left turn is unavailable (Table 2). For these intentionally underspecified objectives, broader search can exploit loopholes in the objective and decoder imperfections. In contrast, greedy search is both more computationally efficient and a conservative default that biases solutions toward more robust, realistic behavior.

See Section A.6 for details and experiments with an additional speed reduction objective.

**Closed-loop planning.** To evaluate latent token search in closed loop we turn to the *nuPlan* dataset and simulator (Caesar et al., 2021). As opposed to WOMD, this dataset includes target route information and proposes a closed-loop simulation challenge based on route following. For our nuPlan experiments, we train a trajectory autoencoder on the nuPlan dataset. Following standard practice, we apply ego history dropout during training (Cheng et al., 2024; Zhang et al., 2025) and include an auxiliary object collision loss. However, in our case no route conditioning is included at training time. Instead, we implement a route-following plan-

---

[2]Successful turning is defined as achieving a cumulative leftward heading change of over $45°$. Before computing this metric, we fit a B-spline to the planner's output to ensure robustness against any potential noise in the model output. We do not include spline fitting as part of the test-time objective used in search.

*Table 4.* **Closed-loop planning results on nuPlan.** We build a closed-loop planner based on latent token search by employing a test-time route-following objective. Evaluated on the `Reduced-Val14` split from Dauner et al. (2023), our planner performs on par with recent imitation learning (IL) baselines despite being trained without any route conditioning.

| Type | Method | CLS-NR↑ | S-CR↑ | S-PR↑ |
|---|---|---|---|---|
| Rule | PDM-Closed (Dauner et al., 2023) | 91.21 | 97.01 | 92.68 |
| RL | Gen-Drive (Huang et al., 2025b) | 87.53 | 95.72 | 89.94 |
| | CarPlanner (Zhang et al., 2025) | 91.45 | 96.38 | 95.37 |
| IL | GameFormer (Huang et al., 2023) | 83.76 | 94.73 | 88.12 |
| | PlanTF (Cheng et al., 2024) | 83.66 | 94.02 | 92.67 |
| | Gen-Drive (Huang et al., 2025b) | 85.12 | 93.65 | 86.64 |
| | Latent token search (ours) | 86.71 | 95.91 | 87.31 |

*Table 5.* **Ablation study on nuPlan.** We first ablate design choices in the search algorithm: search depth (*depth*), beam size (*beam*), and usage of gradient-based latent token optimization after search (*opt.*). We then ablate components in our cost function: the route progress cost (*route*) and the road edge collision cost (*edge*).

| Depth | Beam | Opt. | Route | Edge | CLS-NR↑ | S-CR↑ | S-PR↑ |
|---|---|---|---|---|---|---|---|
| *Search algorithm ablations:* | | | | | | | |
| 3 | 1 | | ✓ | ✓ | 85.35 | 95.28 | 87.09 |
| 3 | 2 | | ✓ | ✓ | 86.12 | 95.60 | 86.79 |
| 3 | 4 | | ✓ | ✓ | 84.97 | 93.71 | 88.66 |
| 4 | 2 | | ✓ | ✓ | 85.44 | 95.60 | 87.38 |
| 3 | 2 | ✓ | ✓ | ✓ | 86.71 | 95.91 | 87.31 |
| *Cost function ablations:* | | | | | | | |
| 3 | 2 | ✓ | ✓ | | 86.27 | 95.60 | 87.43 |
| 3 | 2 | ✓ | | | 78.42 | 95.60 | 71.84 |

ner by leveraging the test-time flexibility enabled by latent token search. In particular, we consider an objective that seeks to maximize progress along the route (measured as arclength traversed along a reference path extracted from nuPlan's route specification) while penalizing excessive deviation from the route.

We find that latent token search with our simple route following objective performs on par with recent imitation learning planners despite being trained without any route conditioning. While recent RL-based approaches can achieve even better performance (Zhang et al., 2025), we see RL and other architectural improvements as orthogonal to our approach, as they can be readily incorporated into our training procedure or decoder architecture in the future. Detailed results on nuPlan's closed-loop benchmark with nonreactive agents, including the final aggregate score (CLS-NR) as well as collision rate with other agents or objects (S-CR) and route progress (S-PR) scores, are presented in Table 4.

**Closed-loop planner ablations.** In Table 5, we vary search depth, beam size and usage of a simple continuous latent token refinement step (details on this step can be found in Section A.7). We find that a search depth of $N = 3$ and a small beam size of two perform best, demonstrating the value of maintaining a large library of maneuvers combined

*Table 6.* **Multi-agent trajectory generation with terminal position guidance.** Compared to baselines on WOMD's interactive validation split, our method achieves the best realism while maintaining centimeter-level adherence to the desired terminal position.

| Method | Gradient-free? | Realism | | Constraint Effectiveness | |
|---|---|---|---|---|---|
| | | $minADE_6 \downarrow$ | $meanADE \downarrow$ | $minFDE_6 \downarrow$ | $meanFDE \downarrow$ |
| Optimization baseline (Jiang et al., 2023a) | No | 4.563 | 5.385 | 0.010 | **0.074** |
| CTG (Zhong et al., 2023) | No | 1.180 | 1.947 | 0.515 | 0.838 |
| MotionDiffuser (Jiang et al., 2023a) | No | 0.533 | 2.194 | **0.007** | 0.747 |
| Consistency model (Li et al., 2025) | No | *0.490* | – | *0.008* | – |
| Latent token search (ours) | **Yes** | 0.496 | *0.767* | 0.399 | 0.974 |
| Latent token search + token opt. (ours) | No | **0.459** | **0.684** | 0.093 | *0.141* |

with a greedy traversal strategy. Consistent with findings that compression is key to the decoder's generative capabilities from the image domain (Lao Beyer et al., 2025), we find that reducing compression leads to reduced performance (see $N = 4$ experiment), and consistent with our prior experiments, we find that greedy search or a small beam size of two works best. We validate our route-following objective by disabling it, which leads to significantly reduced aggregate and route-progress scores. Adding a term to explicitly penalize road edge collisions can lead to a small improvement to the aggregate score. Finally, performing a small number of gradient-based optimization steps with a predictive variance minimization objective yields a further small improvement (note that this optional post-processing step does not require the main route following objective itself to be differentiable).

**Computational efficiency.** With the parameters of $N = 3$, $D = 3$ and $N_l = 2$ used for most experiments in this section, greedy search requires just 24 evaluations of the decoder. On the NVIDIA RTX 6000 Ada GPU, this translates to about 115 trajectories per second, corresponding to about 2760 decoder calls per second. Note also that each call to the environment encoder is amortized across 24 decoder calls during search.

**5.4. Guidance for Multi-Agent Trajectory Generation**

We train a multi-agent trajectory autoencoder with $N = 12$ tokens and token dimensionality $D = 3$. We again impose binary quantization ($N_l = 2$) at test time.

Following MotionDiffuser's evaluation protocol (Jiang et al., 2023a), we consider an objective which minimizes the final displacement error (FDE) defined as the Euclidean distance between each agent's terminal position and the ground truth trajectory's last point. We then validate the effectiveness of latent token search by evaluating realism, as measured by adherence of the generated trajectories to the ground truth trajectories quantified using the average displacement error (ADE) averaged over predicted agents, prediction horizons and object types, following the WOMD convention. Our baselines draw 64 samples for the evaluation of $minADE$ and $meanADE$ metrics. We therefore also generate 64 can-

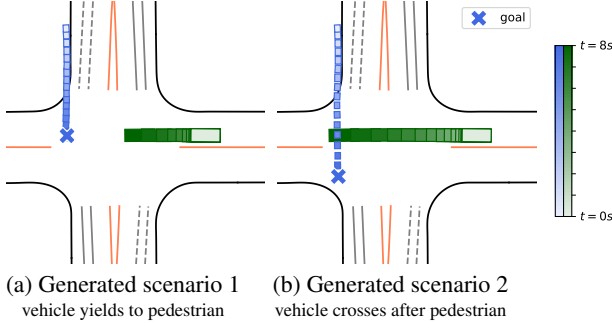

(a) Generated scenario 1
vehicle yields to pedestrian

(b) Generated scenario 2
vehicle crosses after pedestrian

*Figure 6.* **Multi-agent token search generates consistent joint trajectories.** We generate scenarios *(a)* and *(b)* in the same environment by performing token search to minimize the deviation between the final position of the pedestrian (blue) and a user-specified goal point (cross marker). This objective function only supervises the final position of the pedestrian, yet our joint trajectory decoder ensures the behavior of the vehicle (green) is valid.

didates by performing beam search with a beam size of 64, and then evaluate $minADE$, which returns the lowest ADE from among the 6 candidates with the best objective value. For the evaluation of $meanADE$ we generate a single sample using greedy search, since our method is deterministic. The min/mean FDE scores are defined analogously. Since terminal position guidance is an objective that can benefit from micro-adjustments to the final trajectory, we leverage our decoder's ability to handle continuous tokens and optionally include gradient-based optimization of the latent tokens after running search (see Section A.7).

As shown in Table 6, compared to controllable trajectory generation baselines from the literature, our latent token search achieves significantly better realism. The improvement in performance is particularly pronounced when considering mean metrics, highlighting that diffusion methods heavily rely on filtering of samples to achieve good performance, while our latent token search can reliably generate realistic samples without any filtering. As in the single-agent case, selecting the search depth can control the balance between realism and objective maximization, which we explore in Section A.8.

**Qualitative example.** While in the previous experiment we impose guidance on every agent's terminal position, in Figure 6 we guide the terminal goal position *for only a*

*Table 7.* **Effect of token dimensionality on reconstruction and guided generation.** Reconstruction is evaluated using the learned encoder (*AE*), with and without test-time quantization, and using search with a reconstruction objective. Guided generation uses a goal-reaching objective; ADE measures full-trajectory realism, while FDE measures terminal goal error. Quantization and search use binary quantization ($N_l = 2$).

| | Reconstruction ADE | | | Guided generation | |
|---|---|---|---|---|---|
| Dim. | AE cont. | AE quant. | Search | ADE | FDE |
| $D = 2$ | 0.511 | 0.623 | 0.593 | 0.738 | 1.615 |
| $D = 3$ | *0.465* | *0.559* | *0.505* | **0.669** | *1.104* |
| $D = 4$ | **0.462** | **0.539** | **0.417** | *0.736* | **0.757** |

*Table 8.* **Ablation of architectural and training choices on reconstruction and guided generation.** The experimental setup follows that from Table 7.

| | Reconstruction ADE | | | Guided generation | |
|---|---|---|---|---|---|
| Ablation | AE cont. | AE quant. | Search | ADE | FDE |
| Proposed model | *0.465* | *0.559* | *0.505* | **0.669** | *1.104* |
| No causal masking | 0.492 | 0.789 | **0.490** | *0.673* | **1.098** |
| No nested dropout | **0.438** | **0.525** | 1.153 | 1.385 | 4.145 |
| FSQ | 0.681 | 0.684 | 0.569 | 0.724 | 1.395 |

*single agent in the scene*. In this case, the joint trajectory decoder is able to synthesize joint trajectories in which the behavior of *other* agents, whose trajectories do not receive any direct guidance, is consistent.

## 5.5. Representation Learning Ablations

We conclude the experimental evaluation with ablations of the main representation-learning design choices underlying our conditional autoencoder and latent token search. All models in this section are trained for approximately 30% of the number of steps used for the main experiments. We evaluate each model on three tasks: (i) reconstruction using the learned encoder, both with and without test-time quantization; (ii) reconstruction using latent token search with a reconstruction objective; and (iii) guided generation using a goal-reaching objective that minimizes distance to the ground-truth terminal position.

**Token dimensionality.** We first vary the token dimensionality $D$. As shown in Table 7, reconstruction performance improves with increasing token dimensionality, both for the learned encoder and for search with a reconstruction objective. However, the guided generation results in Table 7 show a trade-off between representation capacity and generation quality. While the goal-point error, measured by FDE, improves monotonically with increasing dimensionality, the full-trajectory ADE does not. This suggests that when dense supervision is available, as in reconstruction, higher-capacity tokens improve reconstruction quality. In contrast, when guidance is sparse, as in terminal goal reaching, it is useful to rely more strongly on the decoder by reducing the capacity of its input tokens. In our experiments, $D = 3$ provides the best trade-off between trajectory realism and controllability.

**Architecture and training recipe.** We next ablate the main architectural and training choices: causal masking, nested dropout, and the use of soft quantization during training rather than FSQ (Mentzer et al., 2024). The results in Table 8 show that nested dropout is critical for inducing the desired coarse-to-fine hierarchy among latent tokens. With-

out nested dropout, learned-encoder reconstruction when given the full set of tokens remains strong, but search performance degrades substantially, indicating that the latent space is no longer organized in a way that can be efficiently explored by greedy token search. Causal masking has only a small effect on search performance, although removing it worsens reconstruction under quantization.

We also find that our proposed soft quantization during training, followed by hard quantization at test time, significantly outperforms applying FSQ during training across all metrics including both reconstruction and generation. Furthermore, FSQ appears to reduce the benefit of continuous refinement after search: unlike the soft-quantization setting, removing test-time quantization from the autoencoder does not substantially improve reconstruction.

## 6. Conclusion

We propose a guided generative model in which test-time objectives are optimized by decode-and-score search over an ordered, highly compressed latent code. By making partial token prefixes directly decodable and scorable in trajectory space, this interface supports a large class of test-time objectives without task-specific training, while relying on a learned decoder to enforce realism. We deploy our method in autonomous driving, but hope the underlying idea can find broad applications in controllable generation.

**Limitations and future work.** Our experiments focus on autonomous driving, where trajectories are relatively low-dimensional and strong contextual priors are available from maps and agent histories. An important direction for future work is therefore to extend compressed, ordered trajectory representations to other robotics domains, such as manipulation, which introduce higher-dimensional state and action spaces or contact-rich dynamics. Furthermore, our experiments show success of greedy search for a variety of driving-related test-time objectives. However, future work could include evaluation on tasks with even richer and longer-horizon structure, and explore additional classes of latent search and optimization algorithms.

## Impact Statement

This work supports test-time controllable trajectory generation for robotics. In safety-critical domains (e.g., autonomous driving), responsible deployment requires extensive validation and system-level safeguards beyond the scope of this paper. The methods presented in this work may find additional applications beyond robotics. As with other work whose goal it is to advance the field of machine learning, controllable generation in other domains may have many potential societal consequences, none of which we feel must be specifically highlighted here.

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

# A. Appendix

## A.1. Conditional Autoencoder Model

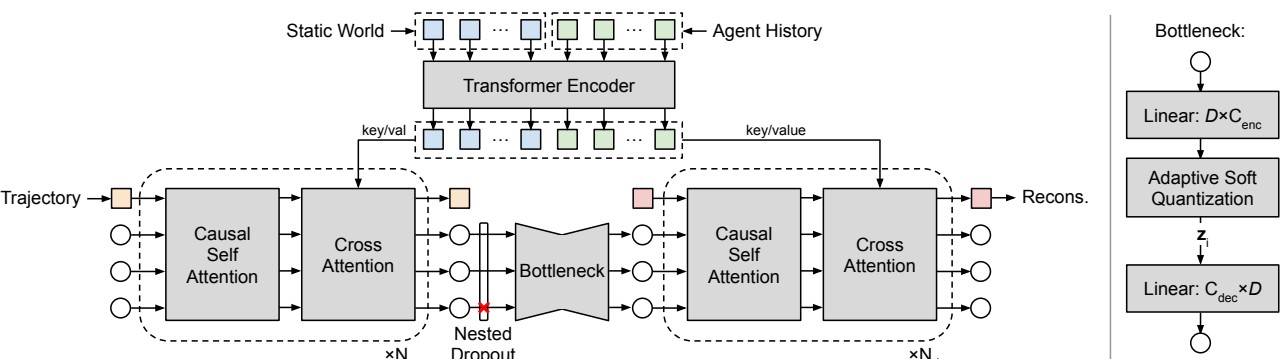

*Figure A1.* **Conditional autoencoder architecture.** The environment (static world, dynamic object histories) is tokenized and fed into a transformer encoder consisting of local neighborhood self-attention layers. The processed tokens form the key and value for the cross-attention-based encoder and decoder transformers. Encoder and decoder impose causal masking among latent tokens.

A graphical overview of the design of our trajectory conditional autoencoder is shown in Figure A1.

**Trajectory and environment tokenization.** We use a similar strategy to MTR (Shi et al., 2022). Agent trajectories are augmented with additional metadata (object type, timestep index, heading angle, and relative position between consecutive samples) and encoded with a PointNet architecture (Qi et al., 2017). Road edges and road lines are split into 20-point segments and also encoded using a small PointNet-style network. In contrast to the original MTR, we encode each feature in the local coordinate frame placed at its initial pose (for agent trajectories) or center of mass (for road geometry).

**Transformer models.** The environment encoder, taking as input the tokenized static world and agent history representations, uses *local self attention* layers to improve efficiency and incorporate a spatial locality bias. We again closely follow MTR (Shi et al., 2022) in this regard.

Our trajectory encoder and decoder make use of blocks of causal self-attention followed by cross-attention. In the case of the encoder, the queries are initialized from the following tokens: (a) the token encoding of the full input trajectory, (b) learnable positional encodings for the latent tokens and (c) one or more additional register tokens (Darcet et al., 2024) (not shown in the diagram in Figure A1). Even though the number of key/value tokens produced by the environment encoder may be large, we do not use any form of local attention or geometric aggregation, since cross-attention with respect to our small number of query tokens is sufficiently fast.

The latent tokens are then processed by the bottleneck consisting of projection to the desired token dimensionality and noise injection (during training) or hard quantization (at test time). After projecting back to the transformer feature dimensionality, the decoder processes the latent tokens with the same design as the encoder, noting that we again include one or more register tokens with learnable positional embedding. Finally, we regress the final trajectory from one of the register tokens using a small MLP.

In both our trajectory encoder and decoder's self-attention layers, causal masking is enforced between the latent tokens, while register and input trajectory tokens may attend to all tokens.

## A.2. Multi-Agent Model Details

As illustrated in Figure A2, the multi-agent version of our conditional trajectory autoencoder consists of cross-attention layers to aggregate environment information (identical to the single-agent case) interleaved with self-attention layers that can aggregate information from *all* trajectories among each other and into the causally masked latent tokens. For invariance to global coordinate transformations, we encode the environment in each agent's local coordinate frame. For invariance to different ordering of the input trajectories, we do not add any positional encoding to the trajectory embeddings fed to the encoder, relying on the transformer's natural permutation equivariance. To also preserve equivariance in the decoder, we again do not add a traditional positional encoding to the input trajectory tokens. Instead, we compute a relative pose embedding for each agent $a$ by first computing the pose of all other modeled agents in $a$'s frame and then processing those poses with a permutation equivariant PointNet-style encoder.

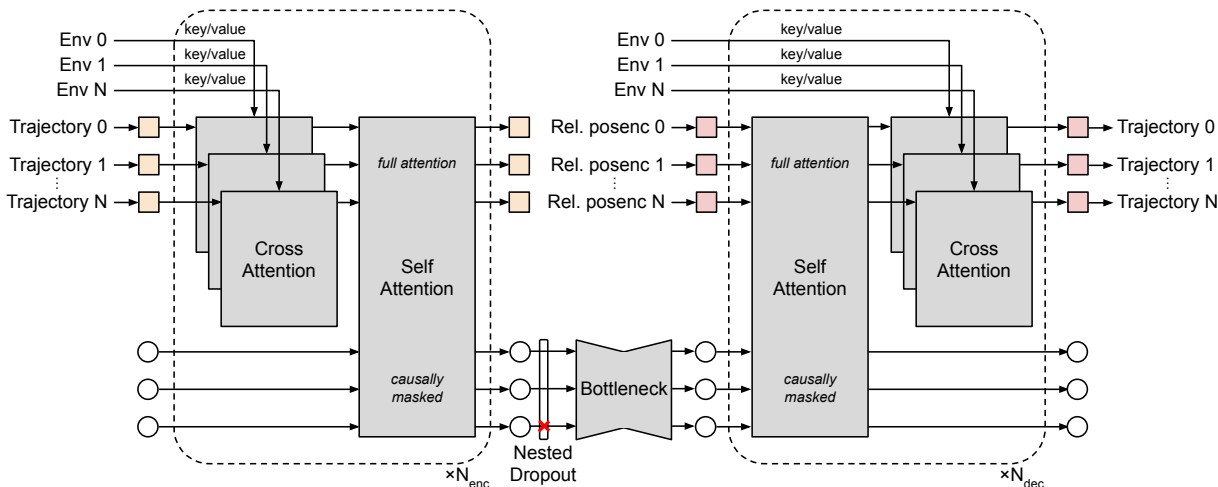

*Figure A2.* **Multi-agent conditional autoencoder architecture.**

## A.3. Model Hyperparameters

The following hyperparameters are shared across the environment encoder model, the trajectory encoder, and the trajectory encoder:

- Layers: 6

- Heads: 8

- Transformer feature dimensionality: 256

- Feed-forward MLP width: 2048

We use these parameters for both the single- and multi-agent experiments.

**Token dropout schedule.** During training, we perform token dropout with $50\%$ probability. If dropout is enabled, we choose the number of tokens to keep according to an exponential schedule with probabilities proportional to $(1/2)^{N_{\text{drop}}}$, where $N_{\text{drop}}$ denotes the number of tokens to drop.

**Other training details.** We train our models using the AdamW optimizer (Loshchilov & Hutter, 2019) with a batch size of 64, learning rate of $10^{-5}$ and weight decay of 0.02. For adaptive soft quantization, we set $\gamma = 0.9995$ and $\Delta\sigma = 0.01$ (see Section A.4). Our single-agent WOMD model is trained on around 70M samples, and we train our nuPlan and multi-agent WOMD models for around 300M samples.

## A.4. Adaptive Noise Injection

During training, we apply a simple elementwise corruption

$$\tanh(\mathbf{z}) + \boldsymbol{\epsilon}_t \quad \text{where} \quad \boldsymbol{\epsilon}_t \sim \mathcal{N}(0, I\sigma_t^2),$$

to each token $\mathbf{z}$ in order to support test-time quantization. We find that choosing $\sigma_t$ to adaptively ramp up significantly improves convergence during training. Concretely, we dynamically choose $\sigma_t$ based on the average displacement error (ADE) averaged across every prediction in the current training minibatch $\text{ADE}_t$:

$$\sigma_t = \max(0, \gamma\sigma_{t-1} + (1-\gamma)\hat{\sigma}_t) \tag{3}$$

$$\text{with} \quad \hat{\sigma}_t = \begin{cases} \sigma_{t-1} + \Delta\sigma & \text{ADE}_t \leq \text{ADE}_{\text{target}} \\ \sigma_{t-1} - \Delta\sigma & \text{ADE}_t > \text{ADE}_{\text{target}} \end{cases}. \tag{4}$$

Here, $\text{ADE}_{\text{target}}$ denotes the target training ADE and is a fixed hyperparameter. Likewise, the decay factor $\gamma \in [0, 1)$ and the noise increment $\Delta\sigma > 0$ can be used to tune the responsiveness of the adaptive schedule to changes in $\text{ADE}_t$.

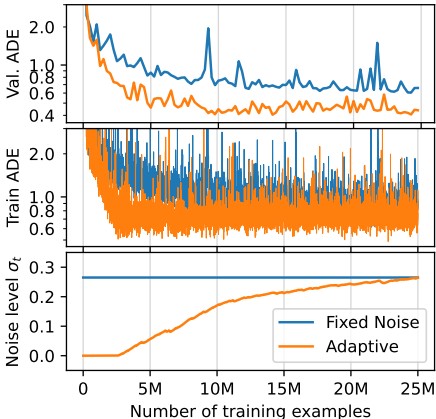

*Figure A3.* **Adaptive noise injection outperforms fixed noise level.** Note that validation ADE is lower than training ADE since $\sigma_t = 0$ during validation.

A unique advantage of this approach is that it yields a decoder that can optionally accept hard-discretized latents at test-time, while retaining the ability to use continuous latents (we exploit this property in Section A.7). Therefore, our diagnostic experiment in Figure A3 does not directly compare against methods using hard quantization such as VQ-VAE (van den Oord et al., 2017) or FSQ (Mentzer et al., 2024).

### A.5. Predictive Uncertainty Calibration

We empirically validate the quality of our model's predictive variance by performing a risk-coverage analysis using 1500 randomly selected WOMD validation scenarios. Our results indicate that the model's variance output is predictive of the reconstruction error and motivates our variance-based token rejection. This holds when not using any test-time quantization as well as when aggressively quantizing to $N_l = 2$, as shown in Table A1. FDE refers to the mean absolute deviation of the final sample of the trajectory.

*Table A1.* **Predictive uncertainty risk coverage analysis.**

| Coverage | FDE (no quant.) | FDE ($N_l = 2$) |
|---|---|---|
| 5% | 0.17 | 0.25 |
| 15% | 0.35 | 0.55 |
| 20% | 0.50 | 0.75 |
| 35% | 0.63 | 0.94 |
| 65% | 0.74 | 1.09 |
| 80% | 0.83 | 1.23 |
| 95% | 0.95 | 1.39 |

### A.6. Details on Planning Objective Functions

Our framework supports general objectives of the form

$$f(\mathcal{T}_{\text{pred}}, z) \mapsto (C, \leq_C), \tag{5}$$

where $\mathcal{T}_{\text{pred}} = \texttt{Dec}(z, \mathcal{E})$ refers to the decoded trajectory, and $z$ is the tokenized representation. In the main text we have considered the left-turn objective with $C = \mathbb{R}$ as well as a lane-change objective with $C = \mathbb{N} \times \mathbb{R}$ under a lexicographic order that first prioritizes the correct number of lane changes and then minimizes a continuous lane centerline deviation.

In our experiments, we consider objectives that penalize high predictive variance, taking the following form. Letting $n$ denote the length of the token sequence $z$ (starting at $n = 1$ at the beginning of search and increasing to $n = N$ once the maximum depth is reached),

$$f_g(\mathcal{T}_{\text{pred}}, z) := g(\mathcal{T}_{\text{pred}}^{(\mu_{xy})}) + \lambda \mathbb{1}[\mathcal{T}_{\text{pred}}^{(\sigma_{xy})} > \sigma_{\max}(n)]. \tag{6}$$

Here, $\mathcal{T}_{\text{pred}}^{(\mu_{xy})}$ refers to the decoder's predicted mean of the trajectory. We use $\mathcal{T}_{\text{pred}}^{(\sigma_{xy})}$ to denote the magnitude of the

predicted covariance for the final sample of the trajectory, which we check against the threshold $\sigma_{\max}(n)$ to impose a heavy penalty $\lambda \gg g(\mathcal{T}_{\text{pred}}^{(\mu_{xy})})$.

We find that an uncertainty threshold which depends on the token sequence length is beneficial, as the distribution of the predicted variance depends strongly on the length of the latent token encoding (see Figure A4). This leaves us to freely choose the cost function $g$ based on the desired target application.

**Left turn maneuver optimization.** The heading $\theta[i]$ along each segment $i$ of the polyline defined by the current candidate trajectory $\mathcal{T}_{\text{pred}}^{(\mu_{xy})}$ is first computed using finite differences. We similarly compute the curvature $\kappa[i]$. We can then compute the total cumulative heading change in the counterclockwise direction as $\text{CCW} = \sum_i \max(0, \theta[i+1] - \theta[i])$. This allows us to define the very straightforward cost function

$$g_{\text{left}}(\mathcal{T}_{\text{pred}}^{(\mu_{xy})}) := -\min\{\text{CCW}, \theta_{\min}\} + \lambda_\kappa \mathbb{1}\{\max_i \kappa[i] > \kappa_{\max}\}, \quad (7)$$

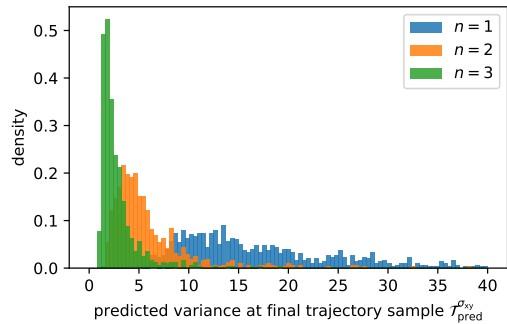

*Figure A4.* **Predicted variance decreases with increasing number of tokens.** Histogram of predicted variance values illustrates need for token-sequence-length-dependent variance threshold during search.

which encourages turns with a leftward heading change of at least $\theta_{\min}$ (which we set to $\frac{\pi}{4}$) while rejecting excessive curvature maneuvers (we set $\kappa_{\max} = 0.35$ very high so that this constraint is rarely active, but $\kappa_{\max}$ could be tuned to match actual vehicle kinematics). In addition to Table 2, we present visualizations of the left turn maneuver optimization in Figure A5 across all scenarios, including those in which a left turn is not available. We highlight that the fact that token search is not always able to achieve the desired heading change is a desirable property: as shown in Figure A5c, our examples include cases where left turns are illegal or impossible.

We also highlight that the ultimate heading change achieved by our token search is *not* simply clustered around the threshold $\theta_{\min}$, indicating that token search finds solutions that align with the correct road geometry rather than blindly optimizing the objective which, if used on its own to optimize a less robust trajectory representation, would be far too simple to produce correct behavior (Figure A5a).

**Speed reduction.** We select $\sim$800 test scenarios in which a vehicle is traveling at an initial speed of around $9\,\text{m/s}$ and maintains a similar average speed throughout its full trajectory. In this case, our objective is to slow down to a lower final speed of $5\,\text{m/s}$ maintained for the last three seconds of the trajectory.

Our choice of $g$ is again very simple:

$$g_{\text{slowdown}}(\mathcal{T}_{\text{pred}}^{(\mu_{xy})}) := \max_{i \in I} \max\{0, v_i - v_{\max}\}, \quad (8)$$

with $v_i$ denoting the magnitude of the velocity along the $i$th segment of the trajectory, again computed using finite differences, $v_{\max}$ denoting the maximum speed constraint value ($5\,\text{m/s}$ in our experiment) and $I$ denoting the range of timesteps to apply the constraint over (in our experiment we apply it from $t = 6\,\text{s}$ to the end of the predicted trajectory at $t = 9\,\text{s}$).

As shown in Table A2, token search does not enjoy a 100% success rate. However, we argue that this behavior is desirable, allowing even very naively specified objective functions to produce reasonable real-world behavior thanks to the decoder serving as a form of "learned guard rails." Indeed, we show in Figure A6c that our framework makes a best effort to increase deceleration while maintaining smooth and safe longitudinal jerk and acceleration values.

*Table A2.* **Speed profile optimization via token search.**

|  | Success Rate | Edge Contact |
|---|---|---|
| None (original scenario) | 0% | 0.76% |
| Token search (1 token) | 28.7% | 0.63% |
| Token search (2 tokens) | 55.4% | 0.38% |
| **Token search (all 3 tokens)** | **63.2%** | **0.13%** |

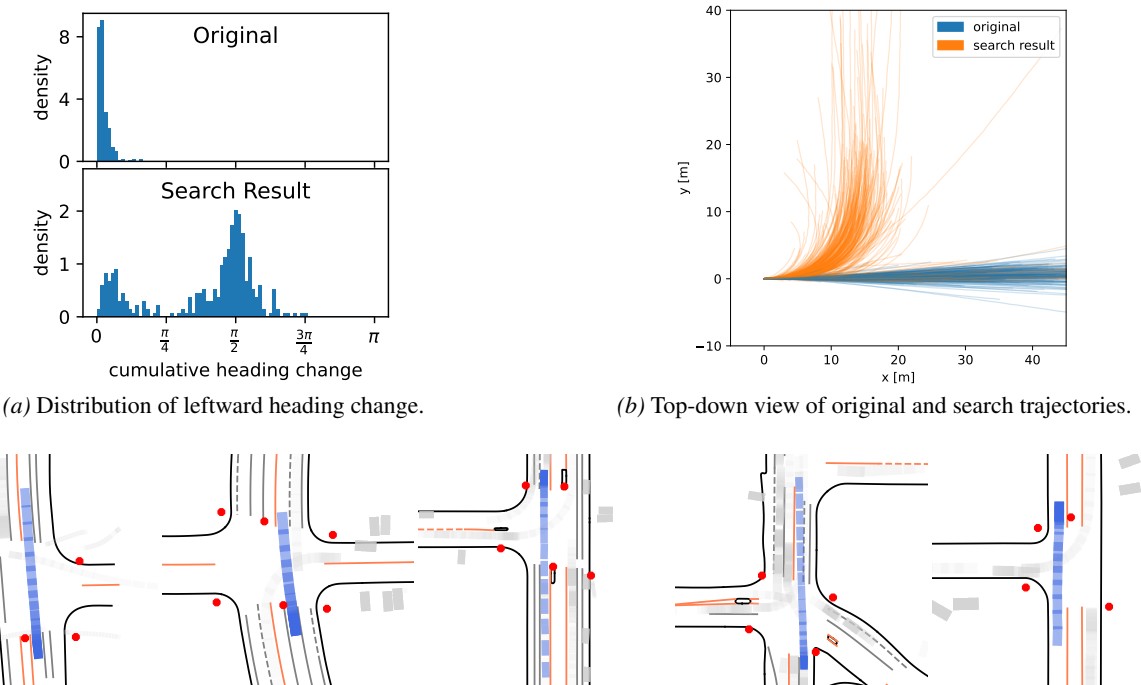

*(a)* Distribution of leftward heading change.

*(b)* Top-down view of original and search trajectories.

*(c)* Typical "failure" cases: left turn would be impossible or illegal (due to not using the dedicated turn lane). We plot the trajectory found by token search with the left turn objective.

*Figure A5.* **Search for left-turn maneuver.** Scenarios are selected by agent's proximity to several stop signs, while moving straight. Some cases where a left turn is not available are included.

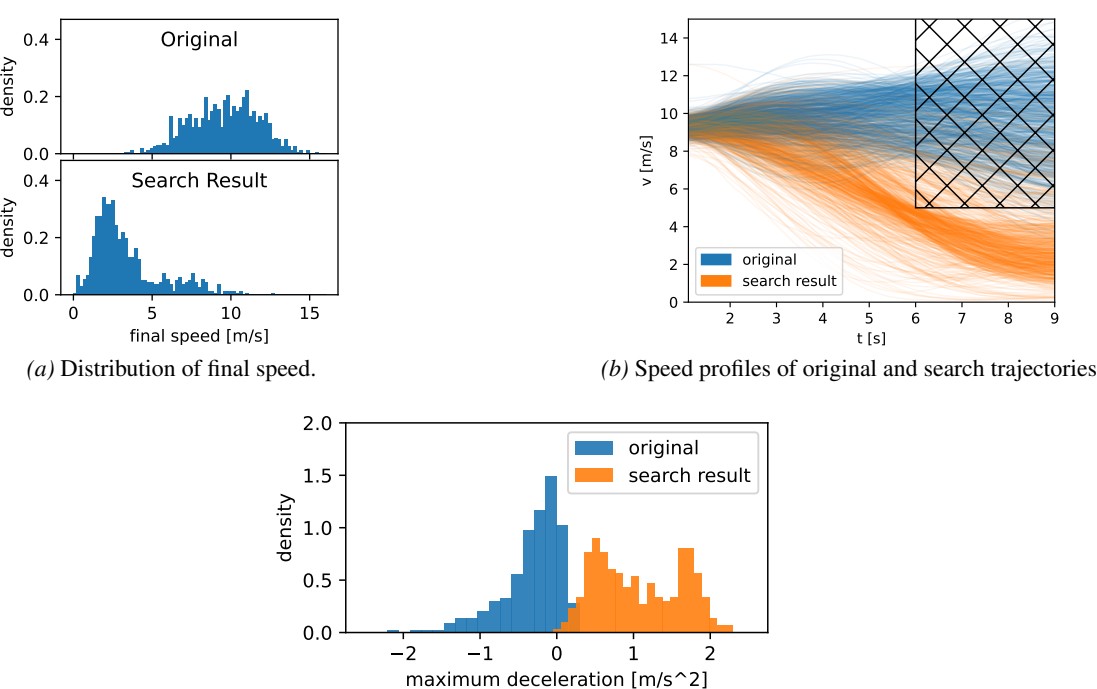

*(a)* Distribution of final speed.

*(b)* Speed profiles of original and search trajectories.

*(c)* Histogram showing maximum *deceleration* of in scenarios where search with speed reduction objective failed to achieve the desired speed reduction. Note that original scenarios are almost exclusively cases with no deceleration or even strong acceleration, and search is able to significantly increase the amount of deceleration.

*Figure A6.* **Slow-down maneuver optimization.** Scenarios are collected by filtering based on the agent's starting speed.

*Table A3.* **Failure of standalone continuous optimization without search.** Standalone optimization is unable to match realism (as indicated by ADE) and objective satisfaction (as measured by FDE) achieved by search or search with optimization as a post-processing step, even when using a larger number of optimizer iterations or initializations. We evaluate on a random 1024-example subset of WOMD's interactive validation split.

| Search | Opt. iter | Num. seeds | ADE↓ | FDE↓ |
|--------|-----------|------------|------|------|
| Yes | 0 | – | 0.793 | 0.948 |
| **Yes** | **40** | **1** | **0.716** | **0.091** |
| No | 40 | 1 | 1.474 | 3.743 |
| No | 120 | 1 | 1.352 | 2.680 |
| No | 40 | 8 | 0.974 | 1.188 |
| No | 120 | 8 | 0.944 | 0.861 |

### A.7. Gradient-Based Token Optimization

In both our closed-loop nuPlan evaluation as well as the multi-agent guided generation experiments we make use of optional gradient-based latent token optimization to boost performance. This simple post-processing step uses the solution found by search as a high quality seed that can be iteratively optimized further according to the same or a different objective. We implement this token optimization via gradient descent, directly backpropagating gradients from the computed objective value to the latent tokens.

In practice, we use Adam with a learning rate of 0.1, optimizing for 5 iterations in the nuPlan experiment (Tables 4 and 5) and for 40 iterations in the multi-agent generation experiment from Table 6.

**Predictive variance minimization.** For the nuPlan experiment (Tables 4 and 5), we choose an objective used during gradient-based optimization that *differs* from the search objective. While search maximizes route-progress, the continuous optimization's objective is to minimize the terminal predicted variance of the solution found by search. This choice leads to a noticeable boost in nuPlan's aggregate closed-loop score without imposing any additional restrictions on the base objective used in search (e.g., it may still be non-differentiable, or use lexicographic ordering of multiple components).

**Final displacement error minimization.** For the multi-agent guided generation experiment from Section 5.4, we use gradient-based optimization to further reduce the same final displacement error already minimized by search. While this optimization does not significantly improve realism metrics, it significantly helps to accurately enforce the prescribed final position. We conclude that gradient-based token optimization is a useful post-processing step in cases where objectives are to be tightly enforced.

**Importance of discrete search prior to optimization.** We find that performing discrete search first is essential in order for the optimization to produce high quality results. To demonstrate this, we compare optimization as refinement of search solutions to standalone optimization with randomly initialized tokens. For this experiment, we use the same multi-agent goal reaching objective as used in Table 6. As illustrated in Table A3, search with no post-optimization beats all of our optimization-only results on realism (as measured by ADE) while being vastly more efficient: searching to a depth of 8 requires only 8 batched sequential calls to the decoder, taking 25 ms per scenario on our hardware (Nvidia RTX 6000 Pro). Meanwhile, the optimization-only method is only able to achieve comparable FDE at significantly worse ADE when selecting the best output among several initializations and running for an extended number of iterations, a configuration which requires 415 ms per scenario (over $16\times$ slower). Search followed by optimization as a post-processing step achieves the best realism and objective satisfaction.

### A.8. Search Depth Selection for Multi-Agent Guided Generation

To explore the dependence of both realism and constraint satisfaction metrics on the search depth, we repeat the experiments from Section 5.4 for a range of possible maximum depth values. For this experiment, we evaluate a random subset of WOMD's interactive validation split containing 1024 scenarios and do not enable gradient-based token optimization.

Figure A7 presents results of our sweep. In all cases, we find that increasing the search depth leads to improved FDE, as expected due to the coarse-to-fine structure of our latent space. However, when using greedy search to generate a single sample per scenario, we observe that ADE (our proxy for realism) stops significantly improving after a depth of just four our of the maximum of $N = 12$ tokens (Figure A7a). When combining this effect with the generation of multiple candidates

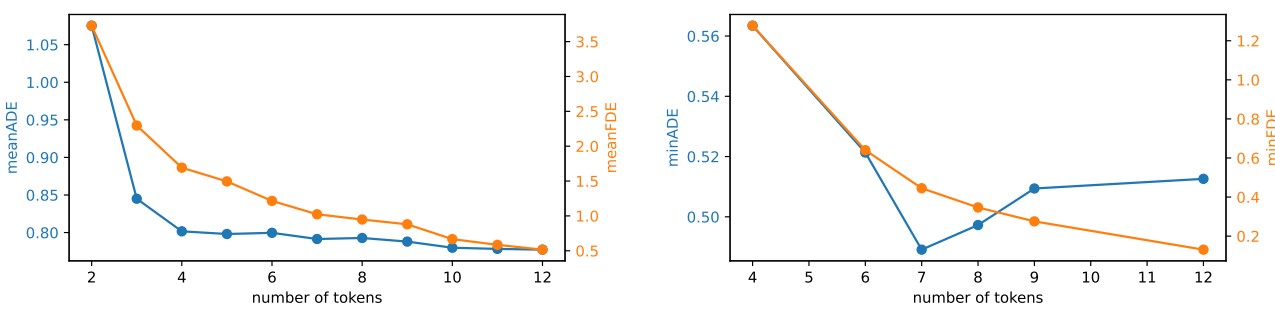

*(a)* Generating a single sample to evaluate mean ADE/FDE     *(b)* Generating multiple samples to evaluate min ADE/FDE

*Figure A7.* **Sweeping the maximum search depth reveals realism/constraint satisfaction tradeoff.**

via beam search, we find that it results in a reduction of realism (Figure A7b). This can be explained by considering the decreasing diversity of candidates available at the end of search at increasing search depths, i.e., at a high search depth, we expect candidates to cluster tightly around an FDE-minimizing solution, but this comes at the expense of reducing coverage and therefore increasing the likelihood of excluding the most realistic solution.

Based on this observation, rather than running search for the full depth of $N = 12$ tokens, our experiments in Table 6 use a depth of $N = 8$ for generation using search only, and $N = 7$ for generation using search and gradient-based optimization.

