# OpenReview forum: "Motion Planning in Compressed Representation Spaces"
_ICML.cc/2026/Conference — ICML 2026 regular_

### Official Review · Reviewer_hgJn · 2026-03-07

**Soundness:** 3
**Presentation:** 4
**Significance:** 2
**Originality:** 3
**Overall Recommendation:** 4
**Confidence:** 2

**Summary:**

The paper proposes a motion planning algorithm that combines learning of generative models from large-scale data and model-based planning. The model compresses the trajectory into a limited set of models, and the method plans by doing search in the latent space of tokens. This search algorithm makes it possible for the algorithm to adapt to any tasks given an arbitrary objective function. The proposed method is evaluated on Waymo Open Motion Dataset and nuPlan, where it achieves competitive performance with similar methods despite no task-specific training.

**Compliance With Llm Reviewing Policy:**

Affirmed.

**Final Justification:**

The rebuttal answered most of my concerns and I will retain my score.

**Key Questions For Authors:**

Questions
1. Table 1 shows that a higher number of discretization levels is better. Why do you use binary quantization in your experiments?

Other than that, see weaknesses above.

**Limitations:**

Limitations could be discussed more explicitly

**Strengths And Weaknesses:**

Strengths
- The paper trains a model that is applicable to a multitude of tasks without any re-training or fine-tuning. It is applicable to unseen tasks at test time.
- The paper relies on the insight that using extreme compression allows separating the controllability and realism of trajectories. The decoder ensures that the trajectories are realistic, whereas the small size of the latent space makes it possible to do efficient search. The Figure 4 experiment where the same token corresponds to similar behavior across environments is very interesting.
- The paper is very well-written, easy to follow, and cites relevant and up-to-date related work.
- The method is computationally very efficient despite using test-time search.
- The ADE and FDE results on nuPlan and WOMD are highly impressive. Despite the naive objective, the results on the left turn tasks are good.
- The method supports non-differentiable objective functions.
- The Figure 6 shows that the tokenization captures the overall dynamics well in a multi-agent setting

Weaknesses
- The transformer models used by the paper are rather small, and the paper does not evaluate the scalability of the proposed method. In particular, if the extremely compressed latent space is compatible with scaling is an open question. Table 5 shows that increasing the depth has a negative impact on the performance.
- One can sense an attempt at broader framing in the paper, starting from the title. However, while the paper achieves impressive results in the driving simulations, it's still unclear if the what the paper proposes is a generic motion planning approach applicable to other domains. The trajectory representation, the network architecture, the compression ratio are all potentially driving-specific design choices.
- One can argue that especially the greedy search is an approach that works particularly well in certain tasks in autonomous driving, where locally optimal tokens are often also globally optimal because the objective functions have monotonic properties. However, there could be tasks in AD, where a greedy objective function fails because getting to the global optimum requires suboptimal moves short-term, such as overtaking. Furthermore, the relatively poor performance on the speed reduction task could be explained by the coarse representation not capturing enough detail about the objective, which leads to greedy search not performing well. Moreover, in other domains than AD, such as manipulation or legged locomotion, greedy might break down.
- Minor: there are some design choices that are not formally ablated over, such as the soft quantization (vs FSQ), nested dropout, and causal masking. The ablations focus on test-time aspects, but not on the representation learning. The design choices do make sense, but validating them would improve the paper, if sufficient computational resources are available.

---

> ### Author Rebuttal · Authors · 2026-03-29
>
> We thank the reviewer for highlighting what we also see as the core contribution: a test-time-plannable learned representation that separates realism (decoder) from controllability (search), and supports unseen non-differentiable objectives without retraining.
>
> In the following, we address your concerns and question.
>
> * **1. Scalability**
>
> We identify two axes of scaling: representation capacity and model size. In the richer joint multi-agent setting, scaling the number of latent tokens is important (Section 5.4 uses up to 12 tokens). Further, guided generation exhibit a tradeoff between decoder prior and guidance strength: high-level objectives such as route following work best with fewer tokens, while dense objectives such as reconstruction benefit from larger capacity. Our approach supports choosing this capacity at test time via truncation. We agree that the paper does not include a dedicated study of encoder/decoder size scaling, which we view as future work.
>
> * **2. Driving-specific design**
>
> We view the framework as generic at the level of "recipe" (environment-conditioned compression, ordered tokens, and decode-and-score search with arbitrary test-time objectives), while the specific architecture, compression ratio, and representation are indeed domain-specific choices. We focus on driving because it offers rich environments and established benchmarks for prediction, planning, and guided generation.
>
> * **3. Greedy search optimality**
>
> We agree that greedy latent search is not expected to be globally optimal for arbitrary objectives or in all domains. We view it as strongest for high-level objectives. For the speed-reduction objective, we manually reviewed 80 of the least successful cases. Over 60% of these scenarios involve the vehicle traversing a large traffic-light-controlled multi-lane intersection at high speed surrounded by dense, fast-moving traffic. An additional 15% correspond to the vehicle traveling in the fast lane or overtaking in dense traffic. In such scenarios, an abrupt deceleration would be unsafe and could lead to stopping within the intersection, so we believe that the low success rate suggests a lack of support of deceleration maneuvers in such contexts in the training data. However, around 6% of the reviewed failing scenarios appeared to allow safe deceleration, and we agree that failures here could be due to a mismatch between the test-time objective and the coarse-to-fine hierarchy as induced by the training-time reconstruction loss.
>
> * **4. Representation learning ablations**
>
> We have added the requested representation learning ablations in our response to Reviewer 2K9P, who also asked for this ablation study.
>
> In summary, we observe the following:
>
> a. The proposed soft-quantization setup significantly outperforms FSQ in all our experiments. That is, training with soft quantization and hard-quantizing at test time also outperforms FSQ.
>
> b. Removing nested dropout substantially degrades search-based generation, as it appears to largely prevent the desired coarse-to-fine hierarchy from being learned.
>
> c. Removing causal masking has a relatively small effect, indicating that the success of greedy search for generation can be achieved mainly through nested dropout.
>
> * **Q1. Why use binary quantization when a finer discretization can achieve better reconstruction results?**
>
> Table 1 uses a dense reconstruction objective that supervises the entire trajectory. In this case, the supervision is perfect, and increasing the discretization levels provides finer-grained control over the decoder output. However, for more high-level objectives (such as the left-turn or route following ones), we do not have such perfect supervision (as we are interested in planning rather than reconstruction), so we would like to heavily rely on the decoder (rather than a precise encoding of a known trajectory) to predict valid trajectories. We find that overly increasing the capacity of the representation by scaling the number of levels can undermine this goal.
>
> To verify this, we repeated the left-turn experiment from Table 2 with a higher number of discretization levels. In the following table, "A" refers to the percentage of trajectories which both successfully turn and are non-colliding, computed over all of the scenarios in which a legal left turn is available. "N" refers to the percentage trajectories that perform a left turn within scenarios in which a left turn is not available.
>
> ||`N_lvl`=2|`N_lvl`=3|
> |:-:|:-:|:-:|
> |Depth|A↑ / N↓|A↑ / N↓|
> |1|68.0% / 0.0%|78.2% / 4.8%|
> |2|86.3% / 0.0%|75.6% / 4.8%|
> |3|89.3% / 2.4%|76.1% / 9.5%|
>
> With a higher number of discretization steps, the success rate (A) is significantly diminished due to a larger number of colliding trajectories, while the "false-positive" rate (N) also significantly increases.
>
> Finer discretization also increases branching factor, so scaling depth is more efficient than scaling discretization levels (or token dimensionality).

---

> > ### Author Rebuttal · Reviewer_hgJn · 2026-04-04
> >
> > Thank you for clarifying the motivation for binary quantization and discussing the limitations of greedy search. I still view the generality and scalability of the approach as weaknesses, so I will keep my current score of weak accept.

---

> > > ### Author Response · Authors · 2026-04-05
> > >
> > > Thank you again for your careful review and valuable suggestions for improvement!
> > >
> > > While we focus on driving, we believe our conditional autoencoder framework can be applied to different types of conditioning modalities such as RGB images, and our paper already provides an training recipe for learning highly compressed, searchable latents. For example, the additional results we have presented in response to your suggested ablation comparing FSQ to the proposed soft quantization demonstrate our method's superiority in the regime of a very low-dimensional bottleneck, and our example from Figure 4 illustrates that context-conditioned compression can effectively learn meaningful and transferable encodings.

---

### Official Review · Reviewer_2K9P · 2026-03-11

**Soundness:** 3
**Presentation:** 2
**Significance:** 3
**Originality:** 4
**Overall Recommendation:** 4
**Confidence:** 2

**Summary:**

This paper introduces a novel generative framework for robotic motion planning that bridges the gap between data-driven learning approaches and classical model-based search methods.

The core method involves training an environment-conditioned autoencoder to compress high-dimensional agent trajectories into a sequence of discrete, low-dimensional tokens. To structure this latent space, the authors employ causal masking and nested dropout during training, which forces the tokens to learn a hierarchical, coarse-to-fine representation of the trajectory. Additionally, the model utilizes an adaptive soft quantization during training, enabling hard discretization of the tokens at test time.

For test-time planning, the framework bypasses the need for gradient-based guidance or task-specific retraining. Instead, it performs a discrete tree search (such as a greedy search) directly over the latent token space. During this search, token candidates are decoded back into physical trajectories and evaluated against user-specified objective functions, which can be custom-defined at test time and non-differentiable.

Contribution:
- Decoupled Planning: The framework divides the workload; a pre-trained decoder ensures realistic movements, while a test-time search focuses on hitting specific goals.
- Structured Tokenizer: The authors built a highly compressed, ordered latent space that acts as an adaptable, learned library of driving maneuvers.
- Proven in Autonomous Driving: The method was successfully tested on major driving benchmarks (Waymo and nuPlan), performing efficiently across multiple complex planning scenarios without requiring task-specific retraining.

**Compliance With Llm Reviewing Policy:**

Affirmed.

**Final Justification:**

My issues are mostly resolved, so I'll keep the score as is.

**Key Questions For Authors:**

My question is similar to the weakness mentioned above. I am not very familiar with this specific research area, so please feel free to point out if there is any misunderstanding. If the authors can adequately address this concern, I would be willing to increase my score.

**Limitations:**

The extreme compression and hard discrete tokenization inevitably limit the model’s ability to enforce highly precise spatial constraints.

**Strengths And Weaknesses:**

Strengths:
1. The paper is clearly written and easy to follow. The motivation is well articulated, highlighting the trade-off between classical model-based planning and learning-based approaches.
2. The authors rigorously evaluate their framework on well-established large-scale driving benchmarks, namely the Waymo Open Motion Dataset and nuPlan, and present thorough empirical analysis.
3. The novelty of the work is reasonable and well-motivated. In particular, the structured trajectory tokenizer cleverly combines an environment-conditioned autoencoder with nested dropout to impose a causal coarse-to-fine hierarchy over trajectory tokens.

Weakness:
1. The related work section is somewhat limited. The paper would benefit from a more comprehensive comparison with the latest developments in generative modeling and planning. This would help readers better assess the novelty and positioning of the contribution.
2. The proposed method shows relatively weaker performance compared to some recent state-of-the-art approaches. On the nuPlan closed-loop benchmark, the latent token search method achieves an aggregate CLS-NR score of 86.71, which is below methods such as PDM-Closed and CarPlanner. This suggests that further improvements may be needed to reach top-tier closed-loop performance.
3. Although the authors provide an ablation study evaluating the influence of search parameters (depth, beam size) and cost functions, some core architectural components require further exploration. Specifically, the influence of the "Nested Dropout" mechanism and the token dimensionality ($D$)  are not explicitly ablated. Since these elements are central to the paper's claims regarding hierarchical ordering and extreme compression, empirical evidence justifying their specific configurations is needed to make the evaluation complete.

---

> ### Author Rebuttal · Authors · 2026-03-29
>
> We would like to thank you for your review and for recognizing the originality of combining ordered and highly compressed latent representations with efficient and flexible test-time search for guided generation.
>
> We agree that the related work section in the current manuscript is lacking an overview of standard learning-based motion prediction and planning methods and will make sure to include this in a camera-ready version.
>
> In the following, we would like to address your remaining questions and concerns.
>
> * **2. Performance on nuPlan compared to state-of-the-art**
>
> Our method is already competitive with IL baselines despite no route conditioning during training, and we believe many improvements incorporated into state-of-the-art approaches such as CarPlanner are orthogonal to our proposed method (e.g., RL-based fine-tuning or more advanced decoder architectures).
>
> * **3. Representation learning ablations**
>
> We agree that these ablations are important in order to justify our proposed design choices. We have therefore performed some additional experiments by training ablated versions of our model. We train the models used in this ablation study for around 30% of the number of steps used for the existing results in the paper.
>
> In summary, we find that nested dropout is critical in inducing the desired coarse-to-fine hierarchy. Without it, latent token search is not successful. We also find that a token dimensionality of 3 gives the best tradeoff between realism of generations and granularity of control achieved by guidance. The full details follow.
>
> We evaluate each model on the following tasks.
>
> a. Reconstruction using the learned encoder, both with and without test-time quantization.
>
> b. Reconstruction using search with a reconstruction objective.
>
> c. Guided generation using a goal-reaching objective that minimizes distance to the ground-truth terminal position.
>
> We begin by trying different token dimensionalities. First, we see that reconstruction performance (task a, labeled as *AE*, and task b, labeled as *search*) follows the expected trend of improving with increasing token dimensionality:
>
> |Token dim.|AE (no quant.)|AE (`N_lvl`=2)|Search ADE (`N_lvl`=2)|
> |:-:|:-:|:-:|:-:|
> |d=2|0.511|0.623|0.593|
> |d=3|0.465|0.559|0.505|
> |d=4|0.462|0.539|0.417|
>
> However, when considering the guided generation task (see table below), we can see that while the goal point error (FDE) monotonically improves with increasing dimensionality, the trajectory realism as quantified by the full-trajectory ADE does not. This result illustrates a trade-off between representation capacity and generation quality: when high-quality, dense supervision is available (such as in the reconstruction case from above), higher capacity tokens lead to clear improvements in reconstruction quality; however, when guidance provided by the objective is less informative (as is the case with the goal point guidance), it is desirable to lean more heavily on the decoder by reducing the capacity of its input tokens. One way this choice can be made within our framework is to shrink the token dimensionality, although we observe a similar phenomenon when scaling the number of discretization levels (details of such an experiment can be found in our response to Reviewer hgJn's question).
>
> |Token dim.|Search ADE (`N_lvl`=2)|Search FDE (`N_lvl`=2)|
> |:-:|:-:|:-:|
> |d=2|0.738|1.615|
> |d=3|0.669|1.104|
> |d=4|0.736|0.757|
>
> Next, we ablate the major architectural and training recipe design decisions including causal masking, nested dropout, and the use of soft quantization compared to FSQ, beginning again with reconstruction tasks (a and b).
>
> We find that nested dropout is critical in inducing the desired coarse-to-fine hierarchy among latent tokens, with causal masking having only a small impact on performance.
> We also find that the proposed soft quantization at training time followed by hard quantization at test-time significantly outperforms FSQ. Furthermore, applying FSQ at training might eliminate the possibility of performing continuous refinement after search (as suggested by the fact that removing the quantization from the autoencoder at test time does not lead to improved reconstructions, in contrast to the soft quantization case).
>
> |Ablation|AE (No quant.)|AE (N_lvl=2)|Search ADE (N_lvl=2)|
> |:-:|:-:|:-:|:-:|
> |proposed model|0.465|0.559|0.505|
> |no causal masking|0.492|0.789|0.490|
> |no nested dropout|0.438|0.525|1.153|
> |FSQ during training|0.681|0.684|0.569|
>
> Our findings from the reconstruction setting also apply to guided generation, as shown in the table below.
>
> |Ablation|Search ADE (N_lvl=2)|Search FDE (N_lvl=2)|
> |:-:|:-:|:-:|
> |proposed model|0.669|1.104|
> |no causal masking|0.673|1.098|
> |no nested dropout|1.385|4.145|
> |FSQ during training|0.724|1.395|
>
> We believe that these additional experiments more explicitly justify the design choices from our paper.

---

> > ### Author Rebuttal · Reviewer_2K9P · 2026-04-02
> >
> > The authors ablated detailed for Q.3. Q.2 is hard to fully solve in the rebuttal period. And I have no additional question. So I prefer to maintain the original score (weak accept).

---

> > > ### Author Response · Authors · 2026-04-05
> > >
> > > We are glad that you found the additional ablations helpful, and thank you for suggesting this improvement to our paper.
> > >
> > > Regarding nuPlan benchmark performance (Q2), our goal is not to outperform the very strongest task-specific model- or RL-based planners, but to demonstrate the high degree of test-time flexibility that latent token search enables while maintaining highly realistic and scene-consistent behavior. We show this in our paper by considering a reconstruction objective, endpoint guidance, high-level maneuver guidance, non-differentiable and lexicographic objectives, in addition to route following. In the case of route following, a simple route-progress objective enables our method to perform competitively with imitation learning planners despite using no task-specific training (i.e., no route conditioning) and supporting a general class of test-time objectives without any retraining.
> > >
> > > We would also like to highlight that on the multi-agent guided generation task from Table 6, which compares against other methods supporting guidance, our approach achieves significantly higher realism, whereas our nuPlan experiments compare against methods which do *not* support test-time guidance.
> > >
> > > Thank you again for your careful review of our paper and valuable feedback!

---

### Official Review · Reviewer_eHV5 · 2026-03-14

**Soundness:** 2
**Presentation:** 3
**Significance:** 2
**Originality:** 3
**Overall Recommendation:** 3
**Confidence:** 4

**Summary:**

This paper proposes a novel method for motion planning within compressed representation spaces. The authors train an environment-conditioned autoencoder that compresses high-dimensional continuous vehicle trajectories into a very small number of causally ordered discrete tokens. This is achieved using adaptive soft quantization and nested dropout techniques during training. During the test phase, the method discards direct trajectory prediction; instead, it performs greedy search or beam search within the discrete token space. It combines the decoder with user-defined objective functions to find optimal trajectories. The paper conducts both open-loop and closed-loop evaluations on the WOMD and nuPlan datasets.

**Compliance With Llm Reviewing Policy:**

Affirmed.

**Key Questions For Authors:**

1.	Under the setting of N=3, D=3, and N_l=2, the model can generate a maximum of only 512 discrete trajectories in a given environment. How do the authors prove that these 512 pre-defined trajectories are sufficient to safely handle long-tail, unstructured, or highly dynamic urban scenarios? Is this fundamentally just an advanced trajectory rollout library?
2.	The core claim of the paper is that discrete token search can replace complex test-time inference procedures. However, in the nuPlan closed-loop test and multi-agent generation experiments, gradient-based token optimization was used to improve scores. Does this not contradict the paper's core contribution of gradient-free search? If the gradient optimization step were completely removed from Table 4, how severely would the nuPlan scores drop?

**Limitations:**

No. Although the authors provide a brief Impact Statement acknowledging the need for system-level safeguards, they completely fail to discuss the fundamental technical limitations of their method in the main text. The authors should frankly discuss: (1) the expressive capacity ceiling caused by the extremely compressed representation space; (2) the inevitability of the greedy search strategy falling into local optima when complex, long-sequence decisions are required; and (3) the fragility of heavily relying on variance penalty hyperparameters to prevent reward hacking. The lack of an in-depth analysis of these technical failure modes is unacceptable.

**Strengths And Weaknesses:**

Strengths
●	Valuable Research Direction: Combining the expressive power of generative models with the controllability of classical tree search to tackle motion planning is a fundamentally interesting and practical approach.
●	Incremental Methodological Innovation: Introducing nested dropout combined with causal masking into trajectory generation to achieve coarse-to-fine decoding demonstrates a good degree of technical innovation.
●	Practical Engineering Design: Using predictive variance as an out-of-distribution (OOD) detection mechanism to dynamically filter out unreasonable trajectories during search is practically useful from an engineering standpoint.
Weaknesses
●	Severely limited representation space and lack of generalization: The paper claims to maintain a large library of maneuvers, but this heavily contradicts its actual parameter settings. In most experiments, the model uses only N=3 tokens, with a token dimensionality of D=3, and is hard-quantized to N_l=2 levels at test time. This means that for any given environment, the planner's entire search space consists of merely 8^3 = 512 possible trajectories. Using a mere 512 discrete trajectories to cover complex, highly dynamic, and long-tail real-world driving environments essentially degrades the system into a very coarse, learned motion primitive library, raising serious safety concerns for complex urban scenarios.
●	Contradictory baseline optimization: The core selling point of the paper is its gradient-free discrete token search. However, in the most critical nuPlan closed-loop planning evaluation and the multi-agent generation experiments, the authors surreptitiously rely on gradient-based continuous latent token optimization as a post-processing step to boost performance. This contradicts and undermines the foundation of their core contribution, exposing the inherent inability of pure discrete search to enforce precise constraints.
●	Marginalization of critical safety mechanisms: Important safety features are deliberately marginalized in the presentation. For instance, the sequence-length-dependent variance penalty is a core mechanism for preventing the search algorithm from exploiting objective function loopholes (reward hacking) to generate absurd trajectories. However, its detailed formulas and heavy reliance on hand-tuned truncation thresholds are hidden in Appendix A.6. The main text avoids discussing the sensitivity of these crucial hyperparameters.
●	Limited practical significance: Forcibly reducing a complex high-dimensional generative problem into a microscopic discrete enumeration problem bypasses the true difficulties of complex space planning. Opting to evade complexity using an extremely restricted bottleneck makes this approach unlikely to have a profound impact on the autonomous driving industry.

---

> ### Author Rebuttal · Authors · 2026-03-29
>
> We thank the reviewer for their detailed feedback and address the concerns below.
>
> * **1. Representation capacity (Question 1).**
>
> In our single-agent experiments, the searched space is indeed small for each given environment (i.e., for each given context consisting of static world and agent history). However, we do not believe that bottleneck cardinality alone fully characterizes the effective capacity of the method. In our framework, the decoder is conditioned on the same information as a standard motion prediction or planning model, while the latent tokens provide an additional low-dimensional input for test-time controllability. Thus, the method's expressive capacity is better understood through the combination of the searched latents (which serve as an interface for controllability) and the context-conditioned decoder (which learns to capture realistic, context-appropriate behavior from data).
>
> We therefore do not view the 512 trajectories as pre-defined, and while they may be viewed as a trajectory library, it is context-dependent rather than fixed. We do not claim to prove safety in all scenarios. Rather, we show on standard autonomous driving benchmarks that a highly compressed representation can support efficient search while maintaining realistic behavior.
>
> * **2. Post-processing via gradient-based optimization (Question 2).**
>
> **nuPlan.** The search-only variant already performs very similarly to the version with continuous refinement: CLS-NR 86.12 vs. 86.71 (see Table 5, second row). Moreover, the continuous step in nuPlan does not optimize the route-following objective used during search, it only minimizes the model's predicted variance. It therefore is an optional post-processing step compatible with any search objective.
>
> **Multi-agent experiments.** In Table 6, we show that realism scores remain state-of-the-art even without any continuous optimization. We also find that search is essential even when a differentiable objective is available. To demonstrate this, we have run a new experiment to compare optimization as refinement of search solutions to optimization of randomly initialized tokens, using the same terminal position guidance from Section 5.4. On a 1024-example subset of WOMD validation_interactive, search-only gives ADE/FDE 0.793/0.948, search followed by 40 optimization steps improves this to 0.716/0.091, while optimization from random initialization performs much worse even with more iterations/seeds (ADE/FDE 0.944/0.861 when using 8 random seeds and picking the best after running 120 iterations). We agree that the role of continuous optimization should be stated more prominently in the main text.
>
> * **3. Predictive variance threshold tuning.**
>
> We agree that the depth-dependence of the variance threshold should be stated more clearly. We therefore ran additional sensitivity experiments on the left-turn task in which we vary the thresholds, setting them based on percentiles of the predicted-variance distribution at each search depth.
>
> Using the thresholds from the experiment in the paper as a baseline, we perturb each threshold individually and report performance using this perturbed set of thresholds.
> Below, "A" refers to the ratio of trajectories which both successfully turn and are non-colliding, computed over all of the scenarios in which a legal left turn is available (note this is slightly different from Table 2, which breaks it down by "turning" and "colliding"). "N" refers to the ratio trajectories that perform a left turn within scenarios in which a left turn is not available. $q$ denotes that the variance threshold for the corresponding token has been set to the $q$-th quantile.
>
> ||thresh. 1|thresh. 2|thresh. 3|
> |:-:|:-:|:-:|:-:|
> |$q$|A↑ / N↓|A↑ / N↓|A↑ / N↓|
> |70%|46.2% / 0.0%|88.0% / 2.4%|89.3% / 2.4%|
> |80%|61.5% / 2.4%|89.3% / 2.4%|88.5% / 4.8%|
> |90%|80.3% / 2.4%|89.7% / 2.4%|87.6% / 4.8%|
> |95%|88.5% / 2.4%|89.3% / 2.4%|88.0% / 4.8%|
> |99%|91.0% / 7.1%|86.3% / 9.5%|87.6% / 2.4%|
> |100%|92.3% / 9.5%|86.3% / 9.5%|87.6% / 2.4%|
>
> We also test a uniform-quantile threshold across all depths:
>
> |q|90%|95%|97%|98%|99%|100%|
> |-|-|-|-|-|-|-|
> |A↑|79.5%|87.2%|88.0%|88.9%|87.6%|90.2%|
> |N↓|4.8%|4.8%|4.8%|7.1%|11.9%|19.1%|
>
> Even with no variance rejection, the model still avoids invalid left turns in most unavailable-left-turn scenes, despite the objective containing no road geometry or safety terms. We view this as evidence that realism and feasibility are supplied primarily by the learned decoder, although variance thresholding does remain an important OOD-detection mechanism.
>
> * **4. Practical significance**
>
> We do not view compression here as evading planning complexity. The decoder is already conditioned on the same static-world and agent-history inputs as a standard trajectory model, while the latent tokens add an extra interface for test-time control. The framework also scales beyond the smallest single-agent setting: in our multi-agent experiments we use up to 12 tokens.

---

### Decision · Program_Chairs · 2026-04-30

**Decision:**

Accept (regular)

**Comment:**

In this paper, the authors propose a motion planning framework that plans directly in a learned compressed latent space rather than in the original trajectory space. An environment-conditioned autoencoder compresses high-dimensional vehicle trajectories into a very small discrete latent space. At test time, instead of predicting a trajectory directly, the framework runs greedy or beam search over this tiny discrete space, decoding each candidate and scoring it against arbitrary user-defined (including non-differentiable) objectives, with optional gradient-based refinement of the continuous latents as post-processing. Because the decoder is conditioned on scene context and trained to produce realistic trajectories, the search itself can focus on satisfying the objective while realism is handled at decoding time, allowing new test-time objectives to be plugged in without retraining or gradient-based guidance. Evaluation covers open and closed loop tasks on Waymo Dataset and nuPlan including reconstruction, route following, left-turn, speed-reduction, and multi-agent guided generation.

Reviews split into two weak accepts (R-2K9P, R-hgJn) and one weak reject (R-eHV5). The positive reviewers highlighted the originality of decoupling realism (decoder) from controllability (search), the strong open-loop ADE/FDE numbers, and the support for non-differentiable objectives without retraining. Their concerns covered missing representation-learning ablations on nested dropout and quantization choice, modest closed-loop nuPlan numbers relative to PDM-Closed and CarPlanner (R-2K9P), and scalability/domain-generality of the extreme-compression design plus greedy-search optimality concerns (R-hgJn). R-eHV5's weak reject rested on four points: the search space (8^3 = 512 trajectories under default settings) is too small for long-tail scenarios; gradient-based refinement in nuPlan/multi-agent settings undermines the gradient-free search claim; variance-threshold tuning is marginalized in the main text; and the framework "evades complexity."

The authors presented a substantive rebuttal. They added the requested ablations: nested dropout is critical (without it, search ADE on goal-reaching degrades greatly); soft quantization at training with hard quantization at test substantially outperforms FSQ; and token dimensionality exhibits a realism/granularity tradeoff (d=3 best for high-level guidance, larger d for dense supervision). For R-eHV5 they made two responses. First, the 512-trajectory framing misses the fact that the decoder is context-conditioned: each code is corresponding to different trajectories in different scenes rather than acting as a fixed  trajectory library. Second, on nuPlan the search-only variant scores 86.12 vs. 86.71 with continuous refinement which directly addresses the claim that gradient optimization was doing most of the work. Other experiments confirmed search itself is essential. R-2K9P and R-hgJn maintained weak accept; R-eHV5 did not engage with the rebuttal.

The technical contribution  - a context-conditioned, highly-compressed latent that enables efficient discrete search with arbitrary test-time objectives; is novel, cleanly motivated, and supported especially with the new ablations. R-eHV5's specific quantitative critique was refuted by the 86.12 vs. 86.71 numbers and the random-init comparison; given no further engagement, the initial objection is on the record but not continued. I recommend acceptance, and encourage the authors to incorporate the new representation-learning ablations, the gradient-refinement attribution numbers, and the variance-threshold sensitivity study into the main text, alongside a more explicit limitations discussion covering greedy-search local optima, scalability, and the domain-specificity of the current architecture.